# GeONet: a neural operator for learning the Wasserstein geodesic

## Abstract

Optimal transport (OT) offers a versatile framework to compare complex data distributions in a geometrically meaningful way. Traditional methods for computing the Wasserstein distance and geodesic between probability measures require mesh-specific domain discretization and suffer from the curse-of-dimensionality. We present *GeONet*, a mesh-invariant deep neural operator network that learns the non-linear mapping from the input pair of initial and terminal distributions to the Wasserstein geodesic connecting the two endpoint distributions. In the offline training stage, GeONet learns the saddle point optimality conditions for the dynamic formulation of the OT problem in the primal and dual spaces that are characterized by a coupled PDE system. The subsequent inference stage is instantaneous and can be deployed for real-time predictions in the online learning setting. We demonstrate that GeONet achieves comparable testing accuracy to the standard OT solvers on simulation examples and the MNIST dataset with considerably reduced inference-stage computational cost by orders of magnitude.

## 1 Introduction

Recent years have seen tremendous progress in statistical and computational optimal transport (OT) as a lens to explore machine learning problems. One prominent example is to use the Wasserstein distance to compare data distributions in a geometrically meaningful way, which has found various applications, such as in generative models (Arjovsky et al., 2017), domain adaptation (Courty et al., 2017) and computational geometry (Solomon et al., 2015). Computing the optimal coupling and the optimal transport map (if it exists) can be expressed in a fluid dynamics formulation with the minimum kinetic energy (Benamou and Brenier, 2000). Such dynamical formulation defines geodesics in the Wasserstein space of probability measures, thus providing richer information for interpolating between data distributions that can be used to design efficient sampling methods from high-dimensional distributions (Finlay et al., 2020). Moreover, the Wasserstein geodesic is also closely related to the optimal control theory (Chen et al., 2021), which has applications in robotics and control systems (Krishnan and Martínez, 2018; Inoue et al., 2021).

Traditional methods for numerically computing the Wasserstein distance and geodesic require domain discretization that is often mesh-dependent (i.e., on regular grids or triangulated domains). Classical solvers such as Hungarian method (Kuhn, 1955), the auction algorithm (Bertsekas and Castanon, 1989), and transportation simplex (Luenberger and Ye, 2015), suffer from the curse-of-dimensionality and scale poorly for even moderately mesh-sized problems (Klatt et al., 2020; Genevay et al., 2016; Benamou and Brenier, 2000). Entropic regularized OT (Cuturi, 2013) and the Sinkhorn algorithm (Sinkhorn, 1964) have been shown to efficiently approximate the OT solutions at low computational cost, handling high-dimensional distributions (Benamou et al., 2015); however, high accuracy is computationally obstructed with a small regularization parameter (Altschuler et al., 2017; Dvurechensky et al., 2018). Recently, machine learning methods to compute the Wasserstein geodesic for a *given* input pair of probability measures have been considered in (Liu et al., 2021; 2023; Pooladian et al., 2023; Tong et al., 2023), as well as *amortized* methods Lacombe et al. (2023); Amos et al. (2023) for generating static OT maps.

A major challenge of using the OT-based techniques is that one needs to recompute the Wasserstein distance and geodesic for new input pair of probability measures. Thus, issues of scalability on large-scale datasets and suitability in the online learning setting are serious concerns for modern

| $t = 0$ | $t \approx 0.33$ | $t \approx 0.66$ | $t = 1$ |

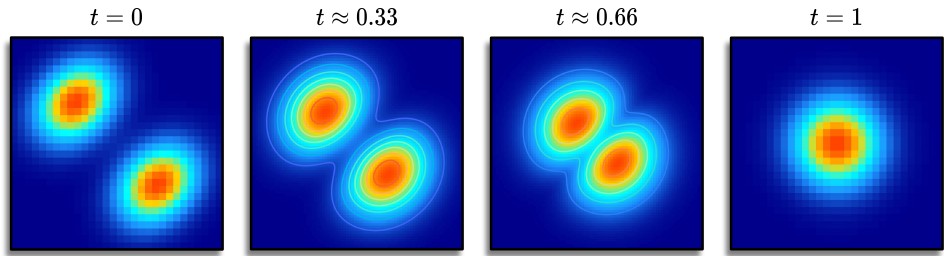

Figure 1: A geodesic at different spatial resolutions. Low-resolution inputs can be adapted into high-resolution geodesics (i.e., super-resolution) with our output mesh-invariant GeONet method.

machine learning, computer graphics, and natural language processing tasks (Genevay et al., 2016; Solomon et al., 2015; Kusner et al., 2015). This motivates us to tackle the problem of learning the Wasserstein geodesic from an *operator learning* perspective.

There is a recent line of work on learning neural operators for solving general differential equations or discovering equations from data, including DeepONet (Lu et al., 2021), Fourier Neural Operators (Li et al., 2020b), and physics-informed neural networks/operators (PINNs) (Raissi et al., 2019) and PINOs (Li et al., 2021). Those methods are mesh-independent, data-driven, and designed to accommodate specific physical laws governed by certain partial differential equations (PDEs).

**Our contributions.** In this paper, we propose a deep neural operator learning framework *GeONet* for the Wasserstein geodesic. Our method is based on learning the optimality conditions in the dynamic formulation of the OT problem, which is characterized by a coupled PDE system in the primal and dual spaces. Our main idea is to recast the learning problem of the Wasserstein geodesic from training data into an operator learning problem for the solution of the PDEs corresponding to the primal and dual OT dynamics. Our method can learn the highly non-linear Wasserstein geodesic operator from a wide collection of training distributions. GeONet is also suitable for zero-shot super-resolution applications on images, i.e., it is trained on lower resolution and predicts at higher resolution without seeing any higher resolution data (Shocher et al., 2018). See Figure 1 for an example of the predicted higher-resolution Wasserstein geodesic connecting two lower-resolution Gaussian mixture distributions by GeONet.

Surprisingly, the training of our GeONet does not require the true geodesic data for connecting the two endpoint distributions. Instead, it only requires the training data as boundary pairs of initial and terminal distributions. The reason that GeONet needs much less input data is because its training process is implicitly informed by the OT dynamics such that the continuity equation in the primal space and Hamilton-Jacobi equation in the dual space must be simultaneously satisfied to ensure zero duality gap. Since the geodesic data are typically difficult to obtain without resorting to some traditional numerical solvers, the *amortized inference* nature of GeONet, where inference on related training pairs can be reused (Gershman and Goodman, 2014), has substantial computational advantage over standard computational OT methods and machine learning methods for computing the geodesic designed for single input pair of distributions (Peyré and Cuturi, 2019; Liu et al., 2021).

Table 1: We compare our method GeONet with other methodology, including traditional neural operators, physics-based neural networks for learning dynamics, and traditional optimal transport solvers.

| Method characteristic | Neural operator w/o physics-informed learning | PINNs | Traditional OT solvers | GeONet (Ours) |
|---|---|---|---|---|
| operator learning | ✓ | | | ✓ |
| satisfies the associated PDEs | ✓ | ✓ | | ✓ |
| does not require known geodesic data | | ✓ | ✓ | ✓ |
| output mesh independence | ✓ | ✓ | | ✓ |

Once GeONet training is complete, the inference stage for predicting the geodesic connecting new initial and terminal data distributions requires only a forward pass of the network, and thus it can be

performed in real time. In contrast, standard OT methods re-compute the Wasserstein distance and geodesic for each new input distribution pair. This is an appealing feature of amortized inference to use a pre-trained GeONet for fast geodesic computation or fine-tuning on a large number of future data distributions. Detailed comparison between our proposed method GeONet with other existing neural operators and networks for learning dynamics from data can be found in Table 1.

## 2 BACKGROUND

### 2.1 OPTIMAL TRANSPORT PROBLEM: STATIC AND DYNAMIC FORMULATIONS

The optimal mass transportation problem, first considered by the French engineer Gaspard Monge, is to find an optimal map $T^*$ for transporting a source distribution $\mu_0$ to a target distribution $\mu_1$ that minimizes some cost function $c : \mathbb{R}^d \times \mathbb{R}^d \to \mathbb{R}$:

$$\min_{T:\mathbb{R}^d \to \mathbb{R}^d} \left\{ \int_{\mathbb{R}^d} c(x, T(x)) \, \mathrm{d}\mu_0(x) : T_\sharp \mu_0 = \mu_1 \right\}, \tag{1}$$

where $T_\sharp \mu$ denotes the pushforward measure defined by $(T_\sharp \mu)(B) = \mu(T^{-1}(B))$ for measurable subset $B \subset \mathbb{R}^d$. In this paper, we focus on the quadratic cost $c(x, y) = \frac{1}{2}\|x - y\|_2^2$. The Monge problem (1) induces a metric, known as the *Wasserstein distance*, on the space $\mathcal{P}_2(\mathbb{R}^d)$ of probability measures on $\mathbb{R}^d$ with finite second moments. In particular, the 2-Wasserstein distance can be expressed in the relaxed Kantorovich form:

$$W_2^2(\mu_0, \mu_1) := \min_{\gamma \in \Gamma(\mu_0, \mu_1)} \left\{ \int_{\mathbb{R}^d \times \mathbb{R}^d} \|x - y\|_2^2 \, \mathrm{d}\gamma(x, y) \right\}, \tag{2}$$

where minimization over $\gamma$ runs over all possible couplings $\Gamma(\mu_0, \mu_1)$ with marginal distributions $\mu_0$ and $\mu_1$. Problem (2) has the dual form (cf. Villani (2003))

$$\frac{1}{2}W_2^2(\mu_0, \mu_1) = \sup_{\varphi \in L^1(\mu_0), \, \psi \in L^1(\mu_1)} \left\{ \int_{\mathbb{R}^d} \varphi \, \mathrm{d}\mu_0 + \int_{\mathbb{R}^d} \psi \, \mathrm{d}\mu_1 : \varphi(x) + \psi(y) \leqslant \frac{\|x - y\|_2^2}{2} \right\}. \tag{3}$$

Problems (1) and (2) are both referred as the *static OT* problem, which has close connection to fluid dynamics. Specifically, the Benamou-Brenier dynamic formulation (Benamou and Brenier, 2000) expresses the Wasserstein distance as a minimal kinetic energy flow problem:

$$\frac{1}{2}W_2^2(\mu_0, \mu_1) = \min_{(\mu, \mathbf{v})} \int_0^1 \int_{\mathbb{R}^d} \frac{1}{2}\|\mathbf{v}(x, t)\|_2^2 \, \mu(x, t) \, \mathrm{d}x \, \mathrm{d}t \tag{4}$$
$$\text{subject to } \partial_t \mu + \mathrm{div}(\mu \mathbf{v}) = 0, \ \mu(\cdot, 0) = \mu_0, \ \mu(\cdot, 1) = \mu_1,$$

where $\mu_t := \mu(\cdot, t)$ is the probability density flow at time $t$ satisfying the continuity equation (CE) constraint $\partial_t \mu + \mathrm{div}(\mu \mathbf{v}) = 0$ that ensures the conservation of unit mass along the flow $(\mu_t)_{t \in [0,1]}$. To solve (4), we apply the Lagrange multiplier method to find the saddle point in the primal and dual variables. In particular, for any flow $\mu_t$ initializing from $\mu_0$ and terminating at $\mu_1$, the Lagrangian function for (4) can be written as

$$\mathcal{L}(\mu, \mathbf{v}, u) = \int_0^1 \int_{\mathbb{R}^d} \left[ \frac{1}{2}\|\mathbf{v}\|_2^2 \mu + (\partial_t \mu + \mathrm{div}(\mu \mathbf{v})) \, u \right] \, \mathrm{d}x \, \mathrm{d}t, \tag{5}$$

where $u := u(x, t)$ is the dual variable for the continuity equation. Using integration-by-parts under suitable decay condition for $\|x\|_2 \to \infty$, we find that the optimal dual variable $u^*$ satisfies the Hamilton-Jacobi (HJ) equation for the dynamic OT problem

$$\partial_t u + \frac{1}{2}\|\nabla u\|_2^2 = 0, \tag{6}$$

and the optimal velocity vector field is given by $\mathbf{v}^*(x, t) = \nabla u^*(x, t)$. Hence, we obtained that the Karush–Kuhn–Tucker (KKT) optimality conditions for (5) are solution $(\mu^*, u^*)$ to the following system of PDEs:

$$\begin{cases} \partial_t \mu + \mathrm{div}(\mu \nabla u) = 0, \ \partial_t u + \dfrac{1}{2}\|\nabla u\|_2^2 = 0, \\ \quad \mu(\cdot, 0) = \mu_0, \ \mu(\cdot, 1) = \mu_1. \end{cases} \tag{7}$$

In addition, solution to the Hamilton-Jacobi equation (6) can be viewed as an interpolation $u(x, t)$ of the Kantorovich potential between the initial and terminal distributions in the sense that $u^*(x, 1) = \psi^*(x)$ and $u^*(x, 0) = -\varphi^*(x)$ (both up to some additive constants), where $\psi^*$ and $\varphi^*$ are the optimal Kantorovich potentials for solving the static dual OT problem (3). A detailed derivation of the primal-dual optimality conditions for the dynamical OT formulation is provided in Appendix B.

## 2.2 Learning neural operators

A neural operator generalizes a neural network that learns a mapping $\Gamma^\dagger : \mathcal{A} \to \mathcal{U}$ between infinite-dimensional function spaces $\mathcal{A}$ and $\mathcal{U}$ (Kovachki et al., 2021; Li et al., 2020a). Typically, $\mathcal{A}$ and $\mathcal{U}$ contain functions defined over a space-time domain $\Omega \times [0, T]$, where $\Omega$ is taken as a subset of $\mathbb{R}^d$, and the mapping of interest $\Gamma^\dagger$ is implicitly defined through certain differential operator. For example, the physics informed neural network (PINN) (Raissi et al., 2019) aims to use a neural network to find a solution to the PDE

$$\partial_t u + \mathcal{D}[u] = 0, \tag{8}$$

given the boundary data $u(\cdot, 0) = u_0$ and $u(\cdot, T) = u_T$, where $\mathcal{D} := \mathcal{D}(a)$ denotes a non-linear differential operator in space that may depend on the input function $a \in \mathcal{A}$. Different from the classical neural network learning paradigm that is purely data-driven, a PINN has less input data (i.e., some randomly sampled data points from the solution $u = \Gamma^\dagger(a)$ and the boundary conditions) since the solution operator $\Gamma^\dagger$ has to obeys the induced physical laws governed by (8). Even though the PINN is mesh-independent, it only learns the solution for a *single* instance of the input function $a$ in the PDE (8). In order to learn the dynamical behavior of the inverse problem $\Gamma^\dagger : \mathcal{A} \to \mathcal{U}$ for an entire family of $\mathcal{A}$, we consider the operator learning perspective.

The idea of using neural networks to approximate any non-linear continuous operator stems from the universal approximation theorem for operators (Chen and Chen, 1995; Lu et al., 2021). In particular, we construct a parametric map by a neural network $\Gamma : \mathcal{A} \times \Theta \to \mathcal{U}$ for a finite-dimensional parameter space $\Theta$ to approximate the true solution operator $\Gamma^\dagger$. In this paper, we adopt the *DeepONet* architecture (Lu et al., 2021), suitable for their ability to learn mappings from pairings of initial input data (Tan and Chen, 2022), to model $\Gamma$. We refer the readers to Appendix F for some basics of DeepONet and its enhanced versions. Then, the neural operator learning problem for finding the optimal $\theta^* \in \Theta$ can be done in the classical risk minimization framework via

$$
\begin{aligned}
\theta^* = \mathrm{argmin}_{\theta \in \Theta} \quad & \mathbb{E}_{(a, u_0, u_T) \sim \mu} \Big[ \big\| (\partial_t + \mathcal{D}) \Gamma(a, \theta) \big\|^2_{L^2(\Omega \times (0, T))} \\
& + \lambda_0 \big\| \Gamma(a, \theta)(\cdot, 0) - u_0 \big\|^2_{L^2(\Omega)} + \lambda_T \big\| \Gamma(a, \theta)(\cdot, T) - u_T \big\|^2_{L^2(\Omega)} \Big],
\end{aligned}
\tag{9}
$$

where the input data $(a, u_0, u_T)$ are sampled from some joint distribution $\mu$. In (9), we minimize the PDE residual corresponding to $\partial_t u + \mathcal{D}[u] = 0$ while constraining the network by imposing boundary conditions. The loss function has weights $\lambda_0, \lambda_T > 0$. Given a finite sample $\{(a^{(i)}, u_0^{(i)}, u_T^{(i)})\}_{i=1}^n$, and data points randomly sampled in the space-time domain $\Omega \times (0, T)$, we may minimize the empirical loss analog of (9) by replacing $\| \cdot \|_{L^2(\Omega \times (0,T))}$ with the discrete $L^2$ norm over domain $\Omega \times (0, T)$. Computation of the exact differential operators $\partial_t$ and $\mathcal{D}$ can be conveniently exploited via automatic differentiation in standard deep learning packages.

## 3 Our method

We present *GeONet*, a geodesic operator network for learning the Wasserstein geodesic $\{\mu_t\}_{t \in [0,1]}$ connecting $\mu_0$ to $\mu_1$ from the distance $W_2(\mu_0, \mu_1)$. Let $\Omega \subset \mathbb{R}^d$ be the spatial domain where the probability measures are supported. For probability measures $\mu_0, \mu_1 \in \mathcal{P}_2(\Omega)$, it is well-known that the constant-speed geodesic $\{\mu_t\}_{t \in [0,1]}$ between $\mu_0$ and $\mu_1$ is an absolutely continuous curve in the metric space $(\mathcal{P}_2(\Omega), W_2)$, which we denote as $\mathrm{AC}(\mathcal{P}_2(\Omega))$. $\mu_t$ solves the kinetic energy minimization problem in (4) (Santambrogio, 2015). Some basic facts on the metric geometry structure of the Wasserstein geodesic and its relation to the fluid dynamic formulation are reviewed and discussed in Appendix C. In this work, our goal is to learn a non-linear operator

$$\Gamma^\dagger : \mathcal{P}_2(\Omega) \times \mathcal{P}_2(\Omega) \to \mathrm{AC}(\mathcal{P}_2(\Omega)), \tag{10}$$

$$(\mu_0, \mu_1) \mapsto \{\mu_t\}_{t \in [0,1]}, \tag{11}$$

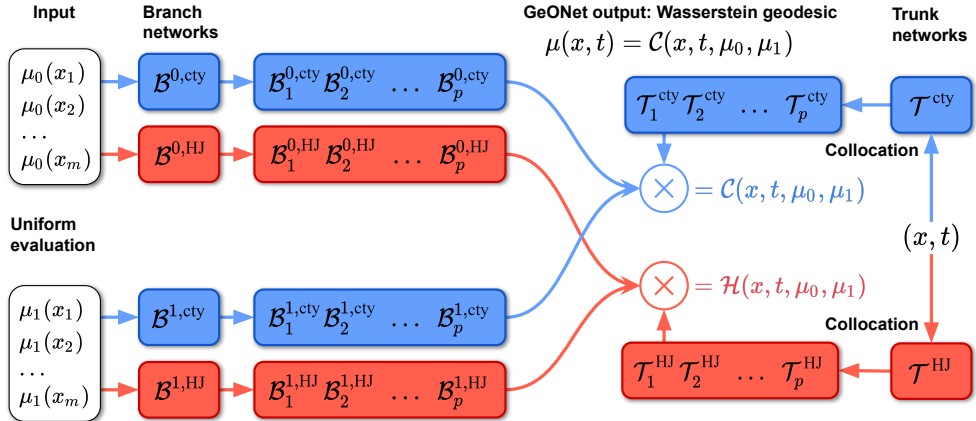

Figure 2: Architecture of GeONet, containing six neural networks to solve the continuity and Hamilton-Jacobi (HJ) equations, three for each. We minimize the total loss, and the continuity solution yields the geodesic. GeONet branches and trunks output vectors of dimension $p$, in which we perform multiplication among neural network elements to produce the continuity and HJ solutions.

based on a training dataset $\{(\mu_0^{(1)}, \mu_1^{(1)}), \ldots, (\mu_0^{(n)}, \mu_1^{(n)})\}$. The core idea of GeONet is to learn the KKT optimality condition (7) for the Benamou-Brenier problem. Since (7) is derived to ensure the zero duality gap between the primal and dual dynamic OT problems, solving the Wasserstein geodesic requires us to introduce two sets of neural networks that train the coupled PDEs simultaneously. Specifically, we model the operator learning problem as an enhanced version of the unstacked DeepONet architecture (Lu et al., 2021; Tan and Chen, 2022) by jointly training three primal networks in (12) and three dual networks in (13) as follows:

$$\mathcal{C}(\mu_0, \mu_1)(x, t, \phi) = \sum_{k=1}^{p} \mathcal{B}_k^{0,\text{cty}}(\mu_0, \theta^{0,\text{cty}}) \cdot \mathcal{B}_k^{1,\text{cty}}(\mu_1, \theta^{1,\text{cty}}) \cdot \mathcal{T}_k^{\text{cty}}(x, t, \xi^{\text{cty}}), \quad (12)$$

$$\mathcal{H}(\mu_0, \mu_1)(x, t, \psi) = \sum_{k=1}^{p} \mathcal{B}_k^{0,\text{HJ}}(\mu_0, \theta^{0,\text{HJ}}) \cdot \mathcal{B}_k^{1,\text{HJ}}(\mu_1, \theta^{1,\text{HJ}}) \cdot \mathcal{T}_k^{\text{HJ}}(x, t, \xi^{\text{HJ}}), \quad (13)$$

where $\mathcal{B}^{j,\text{cty}}(\mu_j(x_1), \ldots, \mu_j(x_m), \theta^{j,\text{cty}}) : \mathbb{R}^m \times \Theta \to \mathbb{R}^p$ and $\mathcal{B}^{j,\text{HJ}}(\mu_j(x_1), \ldots, \mu_j(x_m), \theta^{j,\text{HJ}}) :$ $\mathbb{R}^m \times \Theta \to \mathbb{R}^p$ are *branch* neural networks taking $m$-discretized input of initial and terminal density values at $j = 0$ and $j = 1$ respectively, and $\mathcal{T}^{\text{cty}}(x, t, \xi^{\text{cty}}) : \mathbb{R}^d \times [0, 1] \times \Xi \to \mathbb{R}^p$ and $\mathcal{T}^{\text{HJ}}(x, t, \xi^{\text{HJ}}) : \mathbb{R}^d \times [0, 1] \times \Xi \to \mathbb{R}^p$ are *trunk* neural networks taking spatial and temporal inputs (cf. Appendix F for more details on DeepONet models). Here $\Theta$ and $\Xi$ are finite-dimensional parameter spaces, and $p$ is the output dimension of the branch and truck networks. Denote parameter concatenations $\phi := (\theta^{0,\text{cty}}, \theta^{1,\text{cty}}, \xi^{\text{cty}})$ and $\psi := (\theta^{0,\text{HJ}}, \theta^{1,\text{HJ}}, \xi^{\text{HJ}})$. Then the primal operator network $\mathcal{C}_\phi(x, t, \mu_0, \mu_1) := \mathcal{C}(x, t, \mu_0, \mu_1, \phi)$ for $\phi \in \Theta \times \Theta \times \Xi$ acts as an approximate solution to the continuity equation, hence the true geodesic $\Gamma^\dagger(x, t, \mu_0(x), \mu_1(x)) := \mu_t(x) = \mu(x, t)$, while the dual operator network $\mathcal{H}_\psi(x, t, \mu_0, \mu_1)$ for $\psi \in \Theta \times \Theta \times \Xi$ corresponds to that of the associated Hamilton-Jacobi equation. The architecture of GeONet is shown in Figure 2.

To train the GeONet defined in (12) and (13), we minimize the empirical loss function corresponding to the system of primal-dual PDEs and boundary residuals in (7) over the parameter space $\Theta \times \Theta \times \Xi$:

$$\phi^*, \psi^* = \text{argmin}_{\phi, \psi \in \Theta \times \Theta \times \Xi} \quad \mathcal{L}_{\text{cty}} + \mathcal{L}_{\text{HJ}} + \mathcal{L}_{\text{BC}}, \quad (14)$$

where $\mathcal{L}_{\text{cty}} = \sum_{i=1}^{n} \mathcal{L}_{\text{cty},i}$, $\mathcal{L}_{\text{HJ}} = \sum_{i=1}^{n} \mathcal{L}_{\text{HJ},i}$, $\mathcal{L}_{\text{BC}} = \sum_{i=1}^{n} \mathcal{L}_{\text{BC},i}$, and

$$\mathcal{L}_{\text{cty},i} = \frac{\alpha_1}{N} \| \frac{\partial}{\partial t} \mathcal{C}_{\phi,i}(x, t) + \text{div}(\mathcal{C}_{\phi,i}(x, t) \nabla \mathcal{H}_{\psi,i}(x, t)) \|_{L^2(\Omega \times (0,1))}^2, \quad (15)$$

$$\mathcal{L}_{\text{HJ},i} = \frac{\alpha_2}{N} \| \frac{\partial}{\partial t} \mathcal{H}_{\psi,i}(x, t) + \frac{1}{2} \|\nabla \mathcal{H}_{\psi,i}(x, t)\|_2^2 \|_{L^2(\Omega \times (0,1))}^2, \quad (16)$$

$$\mathcal{L}_{\text{BC},i} = \frac{\beta_0}{N} \| \mathcal{C}_{\phi,i}(x, 0) - \mu_0^{(i)} \|_{L^2(\Omega)}^2 + \frac{\beta_1}{N} \| \mathcal{C}_{\phi,i}(x, 1) - \mu_1^{(i)} \|_{L^2(\Omega)}^2. \quad (17)$$

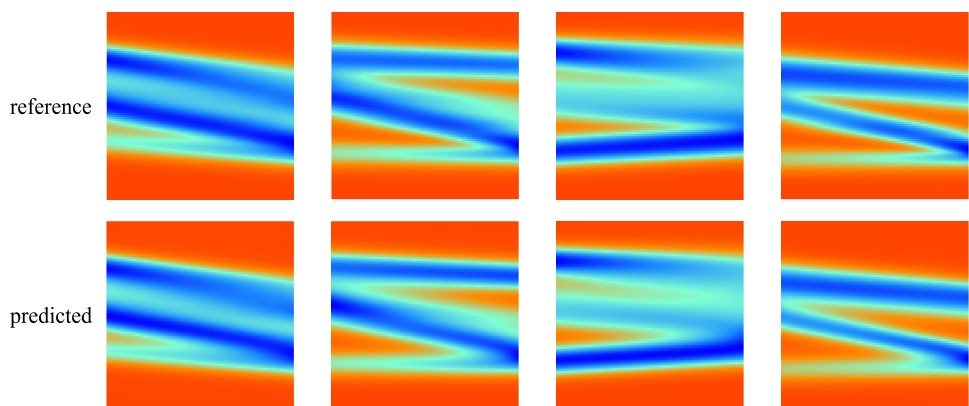

Figure 3: Four geodesics predicted by GeONet with reference geodesics computed by POT on test univariate Gaussian mixture distribution pairs with $k_0 = k_1 = 6$. The reference serves as a close approximation to the true geodesic. The vertical axis is space and the horizontal axis is time.

Here, $\mathcal{C}_{\phi,i}(x,t) := \mathcal{C}_\phi(x,t,\mu_0^{(i)}(x),\mu_1^{(i)}(x))$ and $\mathcal{C}_{\phi,t,i}(x) := \mathcal{C}_\phi(x,t,\mu_0^{(i)}(x),\mu_1^{(i)}(x))$ denote the evaluation of neural network $\mathcal{C}_\phi$ over the $i$-th distribution of the $n$ training data at space location $x$ and time point $t$. The same notation applies for the Hamilton-Jacobi neural networks. $\mathcal{L}_{\text{cty}}$ is the loss component in which the continuity equation is satisfied, and $\mathcal{L}_{\text{HJ}}$ is the Hamilton-Jacobi equation loss component. The boundary conditions are incorporated in the $\mathcal{L}_{\text{BC}}$ term, and $\alpha_1, \alpha_2, \beta_0, \beta_1$ are weights for the strength to impose the physics-informed loss. Automatic differentiation of our GeONet involves differentiating the coupled DeepONet architecture (cf. Figure 2) in order to compute the physics-informed loss terms.

One iterate of our training procedure is as follows. We first select a collection of $N$ indices $\mathcal{I}$ from 1 to $n$ for which of the $n$ distributions are to be evaluated, with possible repeats. For the physics terms, following (Raissi et al., 2019), we utilize a *collocation* procedure as follows. We sample $N$ pairs $(x,t)$ randomly and uniformly within the bounded domain $\Omega \times [0,1]$. These pairs are resampled during each training iteration in our method. Then, we evaluate the continuity and Hamilton-Jacobi residuals displayed in (15) and (16) at such sampled values, in which the loss is subsequently minimized with the corresponding indices in $\mathcal{I}$, making the norms approximated as discrete. For branch input, we take equispaced locations $x_1, \ldots, x_m$ within $\Omega$, a bounded domain in $\mathbb{R}^d$, typically a hypercube $\tilde{\Omega}$. Then the $N$ branch locations are evaluated among $\tilde{\Omega}$ as well for the BC loss.

**Modified multi-layer perceptron (MLP).** A modified MLP architecture as described in (Wang et al., 2021a) has been shown to have great ability in improving performance for PINNs and physics-informed DeepONets. We elaborate on this architecture in Appendix G.1 and describe our empirical findings with this modified MLP for GeONet in Appendix I.

**Entropic regularization.** Our GeONet is compatible with entropic regularization, which is related to the Schrödinger bridge problem and stochastic control (Chen et al., 2016). In Appendix D, we propose the *entropic-regularized GeONet* (ER-GeONet), which learns a similar system of KKT conditions for the optimization as in (7). In the zero-noise limit as the entropic regularization parameter $\varepsilon \downarrow 0$, the solution of the optimal entropic interpolating flow converges to solution of the Benamou-Brenier problem (4) in the sense of the method of vanishing viscosity (Mikami, 2004; Evans, 2010).

## 4 NUMERIC EXPERIMENTS

In this section, we perform simulation studies and a real-data example to demonstrate that GeONet can handle inputs as both continuous densities and discrete point clouds (normalized as empirical probability distributions).

### 4.1 INPUT AS CONTINUOUS DENSITY: GAUSSIAN MIXTURE DISTRIBUTIONS

Since finite mixture distributions are powerful universal approximators for continuous probability density functions (Nguyen et al., 2020), we first deploy GeONet on Gaussian mixture distributions

Table 2: $L^1$ error of GeONet on 50 test data of univariate and bivariate Gaussian mixtures. We compute errors on cases of the identity geodesic, a random pairing in which $\mu_0 \neq \mu_1$, high-resolution random pairings refined to 200 and $75 \times 75$ resolutions in the 1D and 2D cases respectively, and out-of-distribution examples. We report the means and standard deviations as a percentage, making all values multiplied by $10^{-2}$ by those of the table.

| | GeONet $L^1$ error for Gaussian mixtures | | | | |
|---|---|---|---|---|---|
| **Experiment** | $t = 0$ | $t = 0.25$ | $t = 0.5$ | $t = 0.75$ | $t = 1$ |
| 1D identity | $2.67 \pm 0.750$ | $2.85 \pm 0.912$ | $3.04 \pm 1.02$ | $2.86 \pm 0.898$ | $2.63 \pm 0.696$ |
| 1D random | $4.92 \pm 2.00$ | $5.43 \pm 3.02$ | $5.76 \pm 3.56$ | $5.26 \pm 3.25$ | $4.65 \pm 1.50$ |
| 1D high-res. | $4.76 \pm 1.53$ | $5.49 \pm 3.00$ | $6.01 \pm 3.53$ | $5.59 \pm 2.99$ | $4.77 \pm 1.49$ |
| 1D OOD | $12.9 \pm 4.13$ | $14.3 \pm 5.35$ | $16.4 \pm 6.01$ | $14.9 \pm 5.48$ | $12.3 \pm 3.94$ |
| 2D identity | $6.50 \pm 1.15$ | $7.68 \pm 0.915$ | $7.69 \pm 0.924$ | $7.70 \pm 0.889$ | $6.42 \pm 1.11$ |
| 2D random | $6.59 \pm 1.01$ | $7.10 \pm 0.869$ | $7.13 \pm 0.892$ | $7.04 \pm 0.780$ | $6.33 \pm 0.835$ |
| 2D high-res. | $6.66 \pm 0.766$ | $7.71 \pm 1.26$ | $7.88 \pm 1.21$ | $7.59 \pm 0.979$ | $6.29 \pm 0.723$ |
| 2D OOD | $7.15 \pm 0.985$ | $7.82 \pm 1.04$ | $8.14 \pm 1.33$ | $7.96 \pm 1.30$ | $7.14 \pm 0.882$ |

over domains of varying dimensions. We learn the Wasserstein geodesic mapping between two distributions of the form $\mu_j(x) = \sum_{i=1}^{k_j} \pi_i \mathcal{N}(x|u_i, \Sigma_i)$ subject to $\sum_{i=1}^{k_j} \pi_i = 1$, where $j \in \{0, 1\}$ corresponds to initial and terminal distributions $\mu_0, \mu_1$, and $k_j$ denotes the number of components in the mixture. Here $u_i$ and $\Sigma_i$ are the mean vectors and covariance matrices of individual Gaussian components respectively. Due to the space limit, we defer simulation setups, model training details and error metrics to Appendices H, I and J, respectively.

We examine errors in regard to an identity geodesic (i.e., $\mu_0 = \mu_1$), a random test pairing, and an out-of-distribution (OOD) pairing. The mesh-invariant nature of the output of GeONet allows zero-shot super-resolution for adapting low-resolution data into high-resolution geodesics, which includes initial data at $t = 0, 1$, while traditional OT solvers and non-operator learning based methods have no ability to do this, as they are confined to the original mesh. Thus, we also include a random test pairing on higher resolution than training data. The result is reported in Table 2.

## 4.2 INPUT AS POINT CLOUDS: GAUSSIAN MIXTURE DISTRIBUTIONS

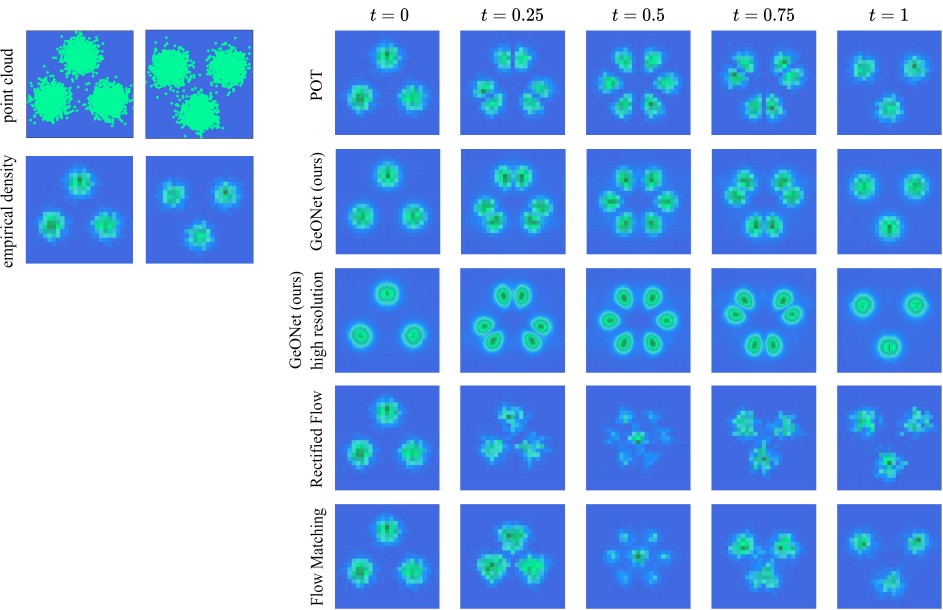

Figure 4: We compare to GeONet to the alternative methodology in a discrete setting, using POT as ground truth. GeONet is the only method among the comparison which encapsulates the geodesic behaviour among the translocation of points.

GeONet is also suited for continuous densities made discrete. In scenarios with access to point clouds of data, we may use GeONet with discrete data made into empirical distributions. We test GeONet an an example Gaussian discrete data. Discrete particles in $\Omega \subseteq \mathbb{R}^2$ are sampled from Gaussian data, as encompassed in (Liu et al., 2022). The sampled particles are represented by empirical densities, in which we compare upon the transition of densities in the non-particle setting using POT as a baseline. The process of turning sampled particles into an empirical density can be reversed by sampling particles according to the densities, which in its most simplified form is placing the particles exactly along the mesh, with the number corresponding to rounded density value. The result is reported in Table 3 and an estimated geodesic example is shown in Figure 4. We observe that RF and CFM have 3-4 times comparably larger estimation errors than GeONet, except for the initial time $t = 0$, because this initial data is given and learned directly for RF and CFM. GeONet is the only framework among the comparison which encapsulates the geodesic behavior to a considerable degree. Second, RF and CFM have the same fixed resolution as the input probability distribution pairing, while GeONet can be smoothed out for estimating the density flows on higher resolution than the input pairing (cf. the third row in Figure 4).

Table 3: $L^1$ error between GeONet, the conditional flow matching (CFM) library's optimal transport solver Tong et al. (2023), and rectified flow (RF) Liu et al. (2022), using POT again as a baseline for comparison. All values are multiplied by $10^{-2}$ to those of the table.

| | $L^1$ comparison error on 2D Gaussian mixture point clouds | | | | |
|---|---|---|---|---|---|
| **Experiment** | $t = 0$ | $t = 0.25$ | $t = 0.5$ | $t = 0.75$ | $t = 1$ |
| GeONet | $22.9 \pm 1.08$ | $28.8 \pm 1.01$ | $30.0 \pm 1.10$ | $29.6 \pm 0.877$ | $22.6 \pm 1.02$ |
| CFM | $0.0 \pm 0.0$ | $94.1 \pm 3.68$ | $98.9 \pm 2.41$ | $91.8 \pm 4.15$ | $75.9 \pm 3.77$ |
| RF | $0.0 \pm 0.0$ | $103 \pm 2.48$ | $112 \pm 3.61$ | $112 \pm 5.03$ | $91.3 \pm 3.79$ |

## 4.3 A REAL DATA APPLICATION

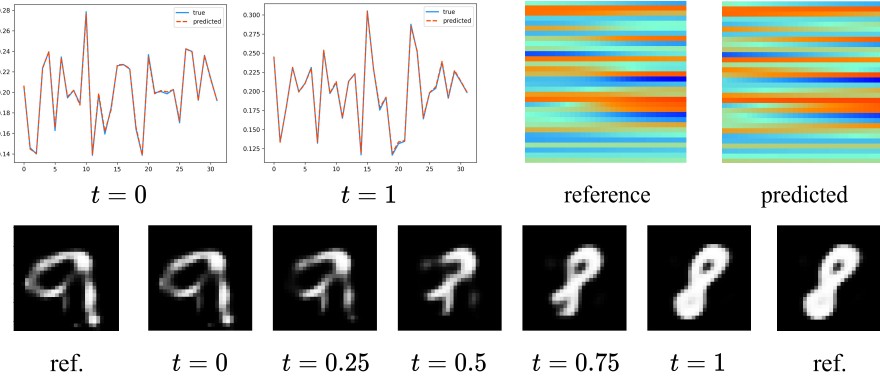

$t = 0 \qquad t = 1 \qquad$ reference $\qquad$ predicted

ref. $\qquad t = 0 \qquad t = 0.25 \qquad t = 0.5 \qquad t = 0.75 \qquad t = 1 \qquad$ ref.

Figure 5: Beginning from top left and going clockwise, we display the initial conditions in the encoded space, the geodesics in the encoded space, and the decoded geodesics as $28 \times 28$ images. (a) and (b) correspond to two unique pairings.

Our next experiment was upon the MNIST dataset of $28 \times 28$ images of single-digit numbers. It is difficult for GeONet to capture the geodesics between digits: MNIST resembles jump-discontinuous data, and relatively piecewise constant otherwise, which is troublesome for the physics-informed term. To remedy our problems with MNIST, we use a pretrained autoencoder to encode the MNIST digits into a low-dimensional representation $v \in \mathbb{R}^{32}$ with an encoder $\Phi$ and a decoder $\Phi^{-1} : v \to \mathbb{R}^{28} \times \mathbb{R}^{28}$ mapping the encoded representation into newly-formed digits resembling that which was fed into the encoder. We institute GeONet upon the encoded representations, learning the geodesic between highly irregular encoded data. Table 4 reports the $L^1$ errors for geodesic estimated in the encoded space and recovered images in the ambient space. As expected, the ambient-space error is much larger than the encoded-space error, meaning that the geodesics in the encoded space and ambient

Table 4: $L^1$ error of GeONet on 50 test pairings of encoded MNIST. All values are multiplied by $10^{-2}$. Error was calculated upon the geodesic in both the shifted and ambient/original space.

| | GeONet $L^1$ error on encoded MNIST data | | | | |
|---|---|---|---|---|---|
| **Test setting** | $t = 0$ | $t = 0.25$ | $t = 0.5$ | $t = 0.75$ | $t = 1$ |
| Encoded, identity | $0.923 \pm 0.213$ | $0.830 \pm 0.166$ | $0.825 \pm 0.165$ | $0.834 \pm 0.173$ | $0.931 \pm 0.215$ |
| Encoded, random | $1.62 \pm 0.333$ | $2.14 \pm 1.22$ | $2.78 \pm 1.62$ | $2.11 \pm 1.17$ | $1.54 \pm 0.282$ |
| Ambient, identity | $26.7 \pm 11.2$ | $34.0 \pm 6.88$ | $35.3 \pm 8.32$ | $36.4 \pm 9.77$ | $34.0 \pm 13.2$ |
| Ambient, random | $32.1 \pm 16.6$ | $58.2 \pm 15.0$ | $68.1 \pm 18.8$ | $56.4 \pm 14.3$ | $24.7 \pm 10.7$ |

image space do not coincide. Figure 5 shows the learnt geodesics in the encoded space and decoded images on the geodesics.

## 4.4 RUNTIME COMPARISON

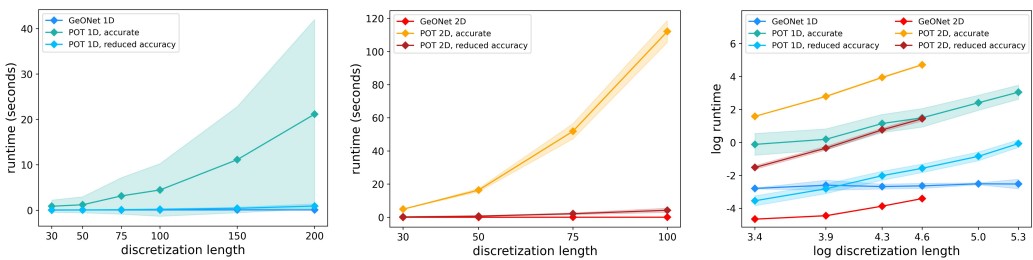

Figure 6: We compare to GeONet to the classical POT library on 1D and 2D Gaussians in terms of mean and standard deviations of runtime on both an unmodified scale as well as one that is log-log using discretization length in one dimension as the x-axis, taken over 30 pairs. We use 20 time steps for 1D and 5 for 2D. Finer meshes are omitted for 2D for computational reasonableness.

Our method is highlighted by the fact that it is near instantaneous: it is highly suitable when many geodesics are needed quickly, or over fine meshes. Traditional optimal transport solvers are greatly encumbered when evaluated over a fine grid, but the output mesh-invariant nature of GeONet bypasses this. In Figure 6, we illustrate GeONet versus POT, a traditional OT library. GeONet greatly outperforms POT for fine grids, especially if POT is used to compute an accurate solution. Even when POT is used to equivalent accuracy, GeONet still outperforms, most illustrated in the log-log plot. The log-log plot also demonstrates that our method speeds computation up to orders of magnitude. We restrict the accuracy of POT by employing a stopping threshold of $0.5$ for 1D and $10.0$ for 2D. We found these choices were comparable to GeONet, remarking a threshold of $10.0$ in the 2D case is sufficiently large so that even larger thresholds have limited effect on error.

## 4.5 LIMITATIONS

There are several limitations we would like to point out. First, GeONet's branch network input exponentially increases in spatial dimension, necessitating extensive input data even in moderately high-dimensional scenarios. One strategy to mitigate this is through leveraging low-dimensional data representations as in the MNIST experiment. While traditional geodesic solvers primarily handle one or two dimensions, GeONet offers a versatile alternative, accommodating any dimension at the cost of potential computational precision. Second, GeONet mandates predetermined evaluation points for branch input, a requisite grounded in the pairing of initial conditions. It is of interest to extend GeONet to include training input data pair on different resolutions. Third, given the regularity of the OT problem (Hütter and Rigollet, 2021; Caffarelli, 1996), developing a generalization error bound for assessing the predictive risk of GeONet is an important future work. Finally, the dynamical OT problem is closely connected to the mean-field planning with an extra interaction term (Fu et al., 2023). It would be interesting to extend the current operator learning perspective to such problems.

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
