# OpenReview forum: "GeONet: a neural operator for learning the Wasserstein geodesic"
_ICLR.cc/2024/Conference — Submitted to ICLR 2024_

### Official Review · Reviewer_oWAG · 2023-10-28

**Soundness:** 3 good
**Presentation:** 3 good
**Contribution:** 3 good
**Rating:** 6
**Confidence:** 5

**Summary:**

The paper introduces a GeONet, a deep learning framework that solves the Optimal Transport problem, which refers to a method for comparing data distributions. Unlike traditional approaches, GeONet is mesh-independent and works well with high-dimensional data. It learns the mapping between input distributions and the Wasserstein geodesic connecting them during offline training, characterized by a PDE system. In the inference stage, it provides real-time predictions with significantly reduced computational cost. GeONet achieves comparable accuracy to traditional methods on various datasets but is much faster during inference.

**Strengths:**

The method applies the PDE method to compute Wasserstein geodesics. It solves a couple of PDE systems involving the continuity equation and the Hamilton-Jacobi equation. The method applies the least squared formulation with neural network parameterizations.

**Weaknesses:**

1. The application of neural networks to compute Wasserstein geodesics is interesting. The accuracy is still a big issue compared to classical mesh-based approaches. See

Fu, et.al., High order computation of optimal transport, mean field planning, and potential mean field games. Journal of computational physics, 2023.

2. Some more interesting computational examples in mean field control problems can be considered in future work. See

Lin, et.al. Alternating the Population and Control Neural Networks to Solve High-Dimensional Stochastic Mean-Field Games, PNAS.

**Questions:**

Some more numerical examples are needed to compare with the current method and the ones studied in the above literature.

---

> ### Author Response · Authors · 2023-11-23
> **Response to Reviewer oWAG**
>
> (1) We agree with you that mesh-based methods are more accurate than learning-based methods. In general, amortized inference are based on reasonable assumptions on data and model (such as no distribution shift in the test time and the model capacity). Mesh-based approaches can accurately solve the Wasserstein geodesic (and other mean-field control/game/planning) down to machine precision (i.e., optimization error), while learning-based methods can only solve up to statistical (coming from data) and approximation (coming from model) errors. On the other hand, learning-based methods are often faster (and sometimes much faster) than mesh-based methods on higher-dimensional and larger-size problems.
>
> (2) We thank you for pointing the interesting direction to make connection to mean-field control problems. The mean-field control problems are very much related to the dynamical OT formulation, both of which can be formulated as a convex-concave primal-dual saddle-point learning (and optimization) problem. On the other hand, there are two main differences. First, initial density and terminal state are required boundary conditions for mean-field control, while Wasserstein geodesic requires the boundary condition as fixed two densities. Second, the dynamical OT problem lacks of the interaction term, which is important in the mean-field control. Thus, it seems that mean-field planning is an intermediate problem which we believe can be studied as an operator learning problem in a similar manner as learning the Wasserstein geodesic. However, learning the dynamics of a general mean-field control is likely harder than dynamical OT problem and we leave it to the future work. We include this point in the *LIMITATIONS* section in the revision.
>
> We remark we have added an additional experiment comparing our methodology to those of alternative literature, on empirical densities constructed from point clouds of Gaussian mixtures. We refer you to section 4.2.

---

### Official Review · Reviewer_Q478 · 2023-10-28

**Soundness:** 3 good
**Presentation:** 2 fair
**Contribution:** 3 good
**Rating:** 6
**Confidence:** 4

**Summary:**

This paper considers the problem of geodestic on probabilistic density space equipped with Wasserstein distance, so that a similar neural operator is trained based on any given two starting density and the ending density to learn the geodesic "curve".  This is based on the Wasserstein distance optimization problem which is replaced with the KKT conditions.  The conditions then are formulated as the objective function to optimize the parametric geodesic curve. It may have certain applications in practice.

**Strengths:**

The paper does demonstrate couple of novel points.

1. It is a novel idea (at least to this reviewer) to convert the "KKT" condition for a variational problem into an objective function for the solution which is based on the Benaniu-Brenier dynamic flow problem and Hamilton-Jacobi equation.

2. Based on the idea used in neural operator, apply the neural operator approach for the PDE defined by the KKT condition.  This is done according to DeepONet architecture

3. Although the topic is quite technical, the presentation is clear, and easy to follow.

**Weaknesses:**

Basically the topic and presentation look excellent, however it lacks of theoretical analysis for example whether the parametric method is appropriate given its complexity of the dynamic geodesic.  Another weak point is it lacks of more examples in application.  See my point in questions.

**Questions:**

1. The design is based on the assumption that we fully know the endpoint densities, however in real-world applications, instead of density, we only have a set of samples for each density.  How does the approach cope with such cases?

2. It is not clear to me how the endpoint densities are defined in MNIST dataset experiment.   I thank you provide the code for the experiment on Gaussian Mixture densities.

---

> ### Author Response · Authors · 2023-11-23
> **Response to Reviewer Q478**
>
> (1) Theoretical analysis. We acknowledge that this paper does not deal with the theoretical aspect of statistical guarantees. We expect that some theoretical work can be done under certain assumptions given the well-known regularity of the (static and dynamic) OT problems. For instance, if the initial and terminal distributions are smooth, then Caffarelli's global regularity ensures that the static OT map between the two distributions is also smooth (to a less degree). Then with reasonable assumptions on the convexity and smoothness of the optimal potential for pushing mass along the Wasserstein geodesic, we expect that there is a stability between the GeONet loss function and the underlying Wasserstein geodesic, leading to a reasonable generalization error bound that is useful to control the predictive risk of geodesic in the test time. Given the experimental nature of this paper introducing a new concept of Wasserstein geodesic from an operator learning perspective, we leave the theoretical analysis to the future work.
>
> (2) We have since remedied this by the introduction of a new experiment on empirical densities constructed from point clouds sampled from Gaussian mixtures. The primary purpose of this experiment is to present a comparison to other methodology, notably rectified flow (RF) [2], and conditional flow matching (CFM) [3]; however, we also remark that this experiment presents GeONet as suitable for a point cloud setting, if point clouds are made into empirical densities. The process can be reversed, by sampling points according to the generated geodesic densities. While GeONet is not a framework for learning the direct translocation of points, we hope this experiment is satisfactory is illustrating effectiveness in the discrete setting.
>
> (3) We thank you for your appreciation of our code. We use an autoencoder framework to encode the entire MNIST data, as a $28 \times 28$ grid made into a $32$-dimensional vector, in which a decoder subsequently maps the encoded representation back to its original image. There are no restrictions on the encoding, and so the encoded representation naturally converges to highly irregular data. These encoded representations are shifted by a constant to ensure nonnegativity, and normalized to be made into densities. The geodesic is then learned between such encoded densities. We revisited this experiment this rebuttal, and ensured the encoded representations were made into densities, which lowered error by several percentage points for all times, in addition to including the full MNIST dataset is used for training. In the revision, we also included the source code for the full MNIST experiments in the supplementary material to reproduce the discretization of the endpoint densities in MNIST data experiment.

---

### Official Review · Reviewer_NBJx · 2023-10-29

**Soundness:** 2 fair
**Presentation:** 3 good
**Contribution:** 1 poor
**Rating:** 3
**Confidence:** 4

**Summary:**

GeONet is a neural operator applied to the atask of amortizing optimal transport geodesics. Specifically, given training pairs of distributions as samples of the density at fixed resolution, a neural operator is learned which produces the geodesic between distributions. This is applied to toy examples such as mixture of gaussians in 1 and 2D as well as to a low dimensional embedding of a subset of MNIST.

**Strengths:**

- Presents a novel framework for amortizing the learning of Wasserstein geodesics based on neural operators.
- The neural operator perspective is interesting in this case.

**Weaknesses:**

- Lack of comparisons and related work:
    - [1,2] both amortize Wasserstein geodesic learning. These should be at least cited and discussed.
    - There are many methods to compute Wasserstein geodesics relatively quickly, although without amortization. I would be curious how the quality of interpolation compares to these more recent methods, and would amend this statement for more recent work [e.g. 3,4,5].

        > Recently, a machine learning method to compute the Wasserstein
        geodesic for a *given* input pair of probability measures has been considered in (Liu et al., 2021).

- The experiments show how well GeONet works in toy settings, although even this is not very clear given the lack of comparisons.
    - The experiments are all extremely toy with limited examples and dimensionality, with the largest experiment being on a small subset (5000) of MNIST on a 30 dimensional embedded space.
    - Why is the ground truth for the Gaussian mixture approximated using Convolutional Wasserstein Barycenters? I’m actually fairly surprised the error is so low, especially for low regularization values.
    - Why is error calculated in the encoded space for MNIST? It would be much more meaningful to calculate error in the ambient space. The error in the encoded space is difficult to understand and an unreproducible metric, particularly given the lack of code.
    - The L_1 error is also twice the Total variation distance. I find it somewhat strange to use TV here, usually Wasserstein or MMD are used, but I guess this is okay for toy problems.
    - Zero shot super resolution can be done by many modern methods. I’m not sure this is a useful experiment without significantly more experimentation and comparison. Perhaps on benchmarks that are constructed with a known map? (see [6]).
    - “x-axis is the log of grid length in one dimension. This is somewhat confusing (also which log base?). Can this be replaced by the actual grid length?
    - While I can see the usefulness of amortizing W2 computation for faster inference, I do not think the comparison in 4.4 is fair. I would be interested to know for a comparable accuracy, how fast POT and GeONet are, as I assume the POT solver is extremely accurate. Or, similar to shown in MetaOT, a comparison showing GeONet provides a better initialization and speeds up convergence of Sinkhorn-based solvers.
    - It could be helpful to include a comparison to non-amortized methods to make clear under what circumstances amortization becomes beneficial.
- The method suffers from the curse of dimensionality as it currently requires fixed sized grids as input. It would be interesting to consider more general input forms.

Overall, while I find this general direction interesting. I find this work underdeveloped, especially when it comes to the experiments. I believe substantially more work is needed for this method to be interesting to an ICLR audience.

1. Julien Lacombe, Julie Digne, Nicolas Courty, and Nicolas Bonneel. Learning to generate wasserstein barycenters, 2021.
2. Brandon Amos, Giulia Luise, Samuel Cohen, and Ievgen Redko. Meta Optimal Transport, ICML 2023.
3. Xingchao Liu, Chengyue Gong, and Qiang Liu. Flow Straight and Fast: Learning to Generate and Transfer Data with Rectified Flow. ICLR 2023.
4. Aram-Alexandre Pooladian, Heli Ben-Hamu, Carles Domingo-Enrich, Brandon Amos, Yaron Lipman, and Ricky Chen. Multisample Flow Matching: Straightening Flows with Minibatch Couplings. ICML 2023.
5. Alexander Tong, Nikoly Malkin, Guillaume Huguet, Yanlei Zhang, Jarrid Rector-Brooks, Kilian Fatras, Guy Wolf, Yoshua Bengio. Improving and Generalizing Flow-Based Generative Models with Minibatch Optimal Transport. 2023.
6. Korotin A, Li L, Genevay A, Solomon JM, Filippov A, Burnaev E. Do neural optimal transport solvers work? a continuous wasserstein-2 benchmark. Advances in Neural Information Processing Systems. 2021 Dec 6;34:14593-605.

**Questions:**

- It is not clear to me how to turn a general 30 dimensional embedding into a distribution in the MNIST example. Could the authors clarify this?

--------------------------------

Edit:

I thank the authors for taking time to respond. I have read and considered the responses and updated manuscript. I believe the manuscript is improved, but still needs more work on the experimental side.

(1b) thank you for the additional comparisons. However, I believe it is much more meaningful to compare against the OT versions of both papers. Specifically, using the re-flow strategy of [3] or the minibatch strategy of [4,5].

(2a) Thank you for including error on the ambient space. The ambient error seems extremely high, even at times zero and one. Perhaps this is not 0-1normalized? I’m not sure how much we can say on this experiment with such a low-quality pre-trained autoencoder even on this small dataset. (Note: I did not have time to read the updated source code).

(2b) I think the entropic regularization parameter used for the ground truth CWB should be noted, and it should be noted that you are actually comparing to an entropic regularized barycenter. I’m not sure if CWB is given the same parameter, but it seems odd to me to report error between an entropic ground truth and a method which is trying to learn the non-entropic barycenter.

(2c) Thank you for including these additional metrics.

(2f) Thank you for the revision of Figure 6. I am more convinced by this, however I would like to ideally see the empirical error against time for these methods, not just a few different settings of the numerical error since a method like GeONet does not really have such a setting. I think this would add context to when GeONet is useful and can provide meaningful amortization.

Overall, I thank the authors for the revision, but more work is needed to fully flush out these experiments. In particular, I would like to see baselines against other OT methods e.g. reflowed RF and minibatch OT CFM. My score remains the same.

---

> ### Author Response · Authors · 2023-11-23
> **Response to Reviewer NBJx**
>
> (1) Comparisons and related work.
>
> (a) Thanks for pointing out the references. In the revised version, we added these two references and made a brief discussion.
>
> (b) In the revision, we added comparison with two more suggested learning-based methods: the rectified flow (RF) [2] and conditional flow matching (CFM) [3]. We run a similar 2D Gaussian mixture simulation setup in the discrete setting, constructing empirical distributions from sampled point clouds belonging from fixed densities. We use POT as the ground truth for comparing GeONet, RF and CFM. The $L^1$ estimation error for geodesic at time point $t = 0, 0.25, 0.5, 0.75, 1$ are reported in Table 3 and an estimated geodesic example is shown in Figure 4. There are a few observations we can draw from this new experiments. First, RF and CFM have 3-4 times comparably larger estimation errors than GeONet, except for the initial time $t = 0$, only because this initial data is given and learned directly for RF and CFM. GeONet is the only framework among the comparison which encapsulates the geodesic behavior to a considerable degree. While the other methods are suitable for point cloud representations, the geodesic behavior is entirely lost and highly inaccurate to ground truth. Second, RF and CFM have the same fixed resolution as the input probability distribution pairing, while GeONet can be smoothed out for estimating the density flows on higher resolution than the input pairing (cf. the third row in Figure 4).
>
> (2) More on experiments and comparisons.
>
> (a) Regarding the MNIST experiment, in the revision, we retrained GeONet on the full MNIST data, splitting with 30,000 training data for $\mu_0$ and 30,000 training data for $\mu_1$ and randomly pairing them together in training. New testing error in the $L^1$ metric is reported in Table 4, consisting of both the error in the encoded space and ambient pixel space. Comparing the encoded-space error in the initial submission (Table 3 therein), we see that increasing the MNIST data indeed decreases the testing errors. Furthermore, the encoded data was also normalized prior to computing the geodesics with GeONet, which likely also contributed to the decrease in error. Moreover, as expected, the ambient-space error is much larger than the encoded-space error, meaning that the geodesics in the encoded space and ambient image space do not coincide. We mentioned this in the *LIMITATIONS* subsection 4.5 (initial submission) that encoding strategy seems to be necessary with such a dataset. In addition, we included the source code for the full MNIST experiments in the supplementary material for our revised submission.
>
> (b) Convolutional Wasserstein Barycenters (CWB) is most suitable for computing the OT maps and dynamics on meshes over large domains in the continuous (non-point cloud) setting, and favorable timing and numerics over other algorithms (linear programs, Sinkhorn, iterative projections) have been reported in [4]. Given the discretization size in our problems, we used off-the-shelf CWB solver in the standard POT Python library to compute the Wasserstein geodesic (since the barycenter is a special of $t = 0.5$). There may be numerical issue when the regularization parameter is smaller than the resolution of the domain discretization because the convolution kernel is ill-conditioned; but similar issues would also occur for other entropic regularized algorithms with low regularization values. In our experiments, it appears that the ``ground truth" geodesics computed by CWB is entirely reasonable (cf. Figures 3, 4, 7 in the revised version). We presume you are referring to the errors between GeONet and the CWB framework. Since CWB acts as the best 2D geodesic calculator in the continuous setting, it is reasonable for comparison to GeONet, yielding such errors.
>
> (c) We choose the $L^1$ error as our primary performance measure because it is a bounded metric. This way, we can interpret this error for different probability distributions across simulation setups. To accommodate your concern, we also include the errors (for the same simulation setups) in terms of $L^2$ distance and Wasserstein distance in Appendix J, Table 6 in the revised version. We see that the $L^2$ and Wasserstein errors exhibits similar patterns as the $L^1$ error, except that it is difficult to interpret the magnitude of the $L^2$ and Wasserstein errors.
>
> (d) We highlight the zero-shot super resolution is consequence of the operator learning (from functions to functions) nature of our method and a feature that is not present in many other existing learned-based [2, 3, 5, 6] and (more traditional) optimization-based Wasserstein geodesic methods. As such, we do not claim the state-of-the-art of super-resolution problems in compute graphics. Given the limited time period for rebuttal, we do not intend to pursue in this direction.

---

> ### Author Response · Authors · 2023-11-23
> **Response to Reviewer NBJx continued**
>
> (e) The log base is the natural logarithm with based $e$. The reason we chose a log-log plot for our runtime comparison is that a log-log plot is capable of displaying a trend when there is a difference in order of magnitude, as exhibited by the linear patterns. According to your suggestion, we added a Figure 7 in Section 4.5 in revision (to replace Figure 7 in Section 4.4 in initial submission) to include plots for runtime vs. discretization length on both non-log and log-log scales.
>
> (f) Thanks for your suggestion! In the revision, we replaced Figure 7 in Section 4.4 by Figure 6 in Section 4.5, which compares the GeONet inference time and two versions of POT: one solves the Wasserstein geodesic problem to the machine precision, and the other solves the same problem with early stopping to the GeONet precision level. The left two panels in Figure 6 (revised version) display the runtime and discretization length (not on the log-scale), and the right panel displays the log-runtime and log-discretization-length to show the order of magnitude differences. The added experiment suggests that: the computational gain of GeONet over both versions of POT is more sizable for higher-dimensional and larger-size problems. The only exception for the reduced accuracy of POT beating GeONet is 1D Gaussian mixture with coarse discretization, a scenario that is not realistically interesting.
>
> (g) We addressed the problem in the reply to a previous question on comparison to non-amortization methods. Specifically, we added comparison with two more suggested learning-based methods: the rectified flow (RF) [2] and conditional flow matching (CFM) [3] in the revision.
>
> (3) Our method only suffers from $\textbf{input}$ curse-of-dimensionality in the branch network, and not $\textbf{output}$ in the trunk network. This suggests that there is no effect on inference time for fine grids of which the output is to be evaluated. Moreover, the input suffers the curse-of-dimensionality only partially: this could be mitigated by using alternative neural network frameworks, such as convolutional neural networks, for the branch networks.
>
> [2] Xingchao Liu, Chengyue Gong, and Qiang Liu. Flow Straight and Fast: Learning to Generate and Transfer Data with Rectified Flow. $\textit{ICLR}$ 2023.
>
> [3] Alexander Tong, Nikoly Malkin, Guillaume Huguet, Yanlei Zhang, Jarrid Rector-Brooks, Kilian Fatras, Guy Wolf, Yoshua Bengio. Improving and Generalizing Flow-Based Generative Models with Minibatch Optimal Transport. 2023.
>
> [4] Solomon et al. (2015) Convolutional wasserstein distances: Efficient optimal transportation on geometric domains. $\textit{ACM Trans. Graph.}$
>
> [5] Julien Lacombe, Julie Digne, Nicolas Courty, and Nicolas Bonneel. Learning to generate Wasserstein barycenters, $\textit{Journal of Mathematical Imaging and Vision}$ 2021.
>
> [6] Brandon Amos, Giulia Luise, Samuel Cohen, and Ievgen Redko. Meta Optimal Transport, $\textit{ICML}$ 2023.

---

### Official Review · Reviewer_1NDe · 2023-10-30

**Soundness:** 2 fair
**Presentation:** 1 poor
**Contribution:** 2 fair
**Rating:** 5
**Confidence:** 3

**Summary:**

This paper is trying to solving the optimal transport optimization to calculate the Wasserstein distance directly with a new neural operator learning.

**Strengths:**

The learned neural operator is extremely fast to calculate the Wasserstein distance.

**Weaknesses:**

I am not sure about whether this net can be generalized to unseen distribution Wasserstein distance calculations especially the ones are very different from the training data. Since in many times we don't know whether the distributions in the application field is similar to the ones in training, and if we just calculate the OT question, we can get the accurate results although it will be slow. But I think it is hard to ensure such a learned neural operator is generally applicable. If need retrain for OOD cases, then I want to know how much data is needed.

Also, very important, the readability is too bad, I think myself is familiar with many maths inside, however the symbols are not defined precisely make it very difficult to understand. I think much more details of the proposed algorithms and backgrounds need to be added. It shall be an important and interesting paper, but if it can not be understood by others then it will be difficult for application fields to use it. Better have more figures about the details as well.

**Questions:**

1. How much data is needed for retraining for OOD cases?
2. I don't quite understand the justification of "no need for retraining for new input", I think it is for applicable for all the usages.
3. I think this paper is rushed, better improve the writing.

I think I will improve the points once a better-written version with the question answered. I think at least the topic is interesting and important, but I don't think it should be presented in this rough way that makes readers very difficult to understand all the details. I will definitely read all the explanations and the revised version of the paper. I expect it will be a much better one.

---

> ### Author Response · Authors · 2023-11-23
> **Response to Review 1NDe**
>
> (1) We thank you for pointing the OOD retraining issue. To clarify, our OOD experiments on 1D and 2D Gaussian mixtures were tested upon data that was not seen in training, but still belonging to the same family of Gaussian mixture distributions with different set of variance parameters from training. Recall that the training dataset size for 1D Gaussian mixtures is 20,000 pairs and for 2D Gaussian mixtures is 5,000 pairs. In the current Gaussian mixture setting, if we were to retrain the experiments on new data with different variance parameters of mixture Gaussian components for GeONet to achieve similar $L^1$ errors, we empirically observe around 500 new pairings are needed.
>
> (2) We apologize for the confusion. This sentence was meant to compare with traditional OT solvers based on optimization (such as iterative Bregman projections [1]) and PINNs that learns the solution of a given (i.e., single) PDE (not the solution operator as a mapping from the boundary conditions to PDE solutions). GeONet is an operator learning method that we do not need to retrain the OT dynamics for new boundary conditions. To accommodate your concern, we changed "no need for retraining for new input" to "operator learning" in Table 1 in the revision.
>
> (3) Thanks for your suggestion. We revised the paper to address questions raised by all reviewers. We hope the rebuttal is further clarifying and that the revised version of the paper is satisfactory.
>
> [1] Benamou, Carlier, Cuturi, Nenna, Peyré. (2015) Iterative Bregman Projections for Regularized Transportation Problems. $\textit{SIAM Journal on Scientific Computing.}$

---

### Meta-Review · Area_Chair_9eDS · 2023-12-06

**Metareview:**

This paper presents a method to compute optimal transport geodesics efficiently by amortization, i.e., by learning a neural operator that generates the geodesic between a pair of distributions, which can then be used on distribution pairs unseen during training. The core of the method involves learning the KKT optimality condition for the Benamou-Brenier problem parametrized through a set of (6) neural networks, which are trained by minimizing an objective on ground-truth (P, Q, Geodesic(P,Q)) triplets. The paper contains simple experimental framework on toy datasets.

Although the reviewers found the paper interesting and well written, they raise concerns about the limited experimental evaluation (restricted to mostly toy datasets), the lack of thorough discussion on some very related work, and some of the assumptions of the method, which they found too restrictive or not sufficiently general. Overall, it seems this paper would benefit from a more robust empirical validation to subsantiate its claims.

**Justification For Why Not Higher Score:**

The experimental evaluation is too limited.

**Justification For Why Not Lower Score:**

N/A

---

### Decision · Program_Chairs · 2024-01-16

Reject